# POINT-IT-OUT: BENCHMARKING EMBODIED REASONING FOR VISION LANGUAGE MODELS IN MULTI-STAGE VISUAL GROUNDING

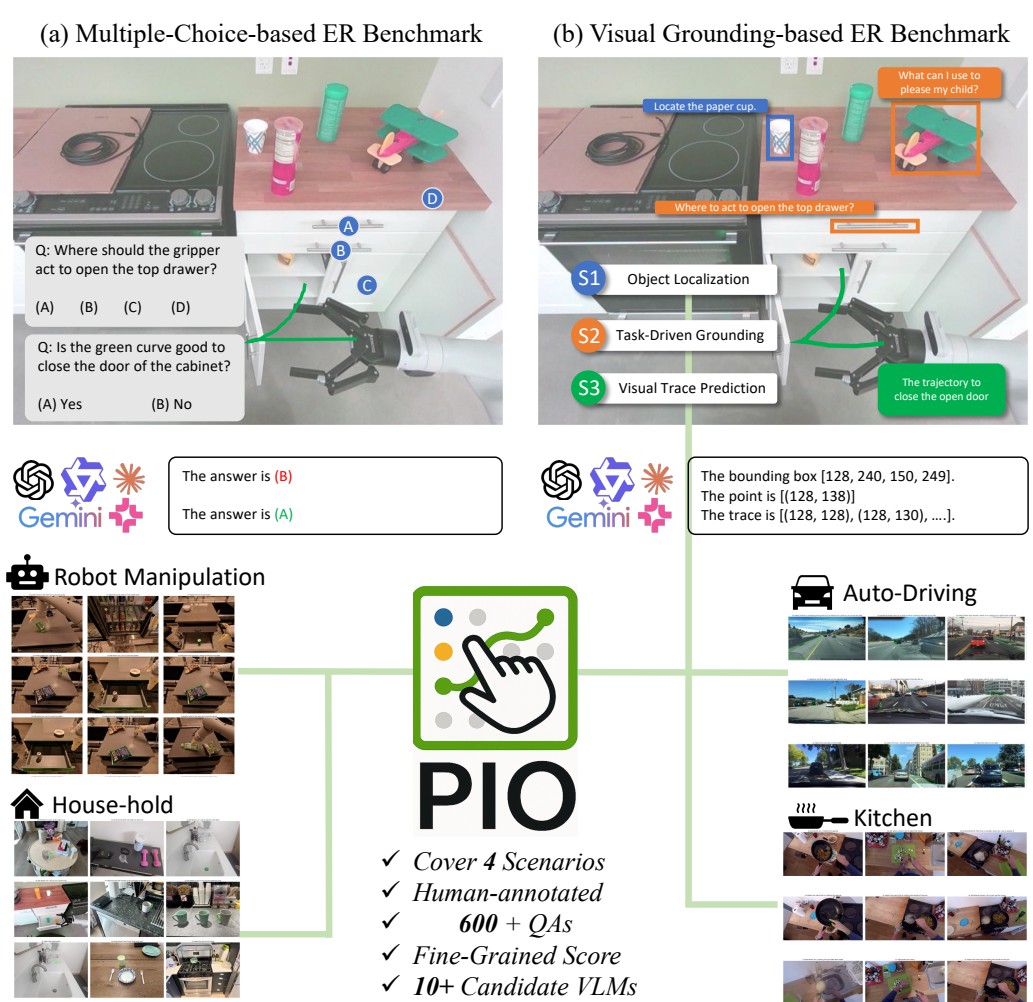

Figure 1: Unlike prior benchmarks that rely on indirect evaluation (a), Point-It-Out (`PIO`) directly assesses embodied reasoning (ER) by prompting VLMs to generate precise visual groundings—such as points, bounding boxes, or trajectories—in a hierarchical manner as shown in (b). To our knowledge, `PIO` is the first benchmark to offer pixel-level grounding for ER, spanning diverse embodied tasks across multiple real-world scenarios.

## ABSTRACT

Vision-Language Models (VLMs) have demonstrated impressive world knowledge across a wide range of tasks, making them promising candidates for embodied reasoning applications. However, existing benchmarks primarily evaluate the embodied reasoning ability of VLMs through multiple-choice questions based on image annotations – for example, selecting which trajectory better describes an

event in the image. In this work, we introduce the Point-It-Out (`PIO`) benchmark, a novel benchmark designed to systematically assess the embodied reasoning abilities of VLMs through precise visual grounding. We propose a hierarchical evaluation protocol spanning three stages (S1: referred-object localization, S2: task-driven pointing, and S3: visual trace prediction), with data collected from critical domains for embodied intelligence, including indoor, kitchen, driving, and robotic manipulation scenarios. Extensive experiments with over ten state-of-the-art VLMs reveal several interesting findings. For example, strong general-purpose models such as GPT-4o, while excelling on many benchmarks (e.g., language, perception, and reasoning), underperform compared to some open-source models in precise visual grounding; models such as MoLMO perform well in S1 and S2 but struggle in S3, where requires grounding combined with visual trace planning.

# 1 INTRODUCTION

Large-scale vision–language models (VLMs) (1; 5; 10; 15; 22) inherit the broad world knowledge and powerful instruction-following abilities of large language models (LLMs) while grounding them in visual inputs. Because these models can describe what they see and reason about how the world works, they have quickly become the backbone of many embodied-AI systems: e.g. robot manipulation (54; 13; 6; 36; 18; 17; 19), navigation (55; 41; 49) and autonomous-driving (38; 50; 31; 50).

Despite the rapid adoption, there are still challenges in understanding the capacities of embodied reasoning (ER) of VLMs, particularly in tasks requiring fine-grained visual grounding. Existing benchmarks primarily focus on input-side understanding and perception, typically using usually evaluate models with either multiple-choice questions (MCQs), e.g., "Which of these trajectories reaches the mug?" (12; 43; 4), or closed-set skill selection from predefined actions (48; 27), or language based planning (48; 2; 56). They either assume that the correct answers are in a list of choices or only provide language-based planning. However, they overlook the crucial step of **grounding** the outputs back into the visual space, which completes the perception–action loop. Without this visual grounding, it is difficult to assess whether a model can truly reason and act in the physical world. Precise visual grounding is therefore essential for evaluating embodied reasoning in a realistic and interpretable manner. Such MCQs and language-based evaluation fail to examine the VLM's capability for fine-grained visual grounding and precise planning, which is critical for ER.

> **Claim**: Current Embodied Reasoning benchmarks (Table 1) offer partial insights by focusing on grounded inputs or language-based planning, but they overlook the need for *precise pixel-level grounding*— a crucial step for making VLMs interpretable and actionable interfaces in real-world embodied tasks (Section 3).

To bridge this gap, we propose to include *visual grounding* (52; 33; 35) as a natural complement to language-based planning in embodied reasoning benchmarks. Here, we adapt the definition of *visual grounding* from (45) into embodied reasoning tasks: by prompting models to localize pixel-space bounding boxes, points, or trajectories based on language-described tasks, we directly assess their accuracy against ground-truth human annotations, providing a clear measure of their embodied reasoning capabilities under precise visual grounding settings. In this paper, we focus on 2D pixel coordinates because precise 2D visual grounding is a scalable, cost-effective proxy task that isolates core embodied reasoning from control dynamics, enabling efficient evaluation.

We propose `PIO`, a benchmark designed to systematically evaluate VLMs' embodied reasoning through precise visual grounding tasks across diverse real-world settings. `PIO` employs a hierarchical evaluation protocol that decomposes embodied reasoning into three stages of increasing complexity: (S1) referred object localization, (S2) task-driven pointing, and (S3) visual trace prediction for spatiotemporal grounding. This structure mirrors the natural complexity of embodied tasks progressively from simple object detection to more challenging tasks such as affordance prediction, spatial reasoning, and task understanding. We further divide S1 and S2 into finer sub-categories, with all labels annotated by humans, providing rich signals for assessing the ER capabilities of VLMs.

Our benchmark includes data from four key domains critical for embodied intelligence: household rooms, kitchen environments, driving scenes, and robotic manipulation tasks. These scenarios require

Table 1: **Comparison of `PIO` with Existing Embodied Reasoning Benchmarks**: We compare benchmarks across five dimensions: (i) the range of *scenarios* covered, where icons denote robot manipulation ⬚, household environments ⌂, kitchens ⬚⬚, and driving scenes ⬚; (ii) the total number of *tasks or questions*; (iii) whether the benchmark requires *pixel-grounded outputs* (e.g., bounding boxes or keypoints); (iv) the presence of *multi-modal input* (e.g., vision and language); and (v) the *question type* or expected model output format. While prior work predominantly focuses on language-based or multiple-choice evaluation formats, `PIO` provides fine-grained, human-annotated pixel-level signal across diverse embodied domains and task types (S1, S2, S3; see Section 3).

| Benchmark | Scenarios | # Tasks | Pixel Grounded | Multi-modality | Question Type |
|---|---|---|---|---|---|
| Cosmos-Reason1 (4) | ⬚⬚ | 600 | ✗ | ✓ | Mutiple-Choices or T/F |
| Gemini-ERQA (43) | ⬚ | 400 | ✗ | ✓ | Mutiple-Choices |
| EmbodiedBench (48) | ⬚(sim) | 1200 | ✗ | ✓ | Mutiple-Choices |
| EmbAgentInterface (27) | ⬚(sim) | 438 | ✗ | ✗ | Language Plan |
| EmbSpatial-Bench (12) | ⌂ | 3640 | ✗ | ✓ | Language Plan & Skill Choose |
| Where2Place (54) | ⌂ | 231 | points | ✓ | Vacant Space Placement |
| RoboRefIt (32) | ⌂ | 10k | bbox | ✓ | Location |
| RefSpatial-Bench (58) | ⌂ | 200 | points | ✓ | Location / Placement |
| `PIO` (ours) | ⬚⌂⬚⬚⬚ | 600 | points,bbox,visual trace | ✓ | S1,S2,S3 (Section 3) |

varying degrees of perceptual grounding, object understanding, spatial navigation, and physical interaction, which are core capabilities for any vision-language agent operating in the real world.

We conduct extensive experiments across a wide range of state-of-the-art VLMs, including general VLM e.g. GPT-4o (1), Claude-3.7 (3), Gemini 2.0-flash (22), Qwen2.5-VL (5), MoLMo-7B (10);some strong reasoning models such as GPT-o3 (37) and Gemini-2.5 (22). Also, we test models that are specifically fine-tuned on grounding tasks e.g RoboRefer (58) and MolmoAct (26). Models explicitly trained for grounding tasks, such as Roborefer, Qwen2.5-VL and Molmo, consistently outperform more general-purpose VLMs, including GPT-o3 and Claude-3.7. For all models, our results reveal that there are still large performance gaps in precise visual grounding within embodied reasoning settings, particularly in tasks requiring fine-grained localization and reasoning about object affordances or physical contact.

In conclusion, our contributions are listed as follows:

1. We introduce **precise visual grounding** as a critical and scalable proxy for embodied reasoning, addressing the limitations of existing benchmarks that primarily rely on multiple-choice evaluations (Table 1).

2. We introduce `PIO`, a three-stage hierarchical benchmark (Section 3) spanning referred-object localization, task-driven pointing, and visual trace prediction. The benchmark includes over 600+ human-annotated datapoints across diverse embodied scenarios (Section 4).

3. We evaluate over ten candidate vision-language models (VLMs) and uncovering key limitations in their precise visual grounding capabilities for embodied reasoning. Our findings highlight the need for targeted data to improve model grounding-aware capabilities (Section 5).

## 2 BACKGROUND AND RELATED WORKS

### 2.1 VISUAL GROUNDING OF VISION-LANGUAGE MODELS

Before large-scale Vision–Language Models (VLMs) emerged, visual grounding research is focused on *referring-expression comprehension* (REC) (47; 52; 33). Pioneering datasets such as ReferItGame (23) and RefCOCO (52) framed the task as localizing the image region that matches a unstructured language description. Modern VLMs have pushed REC performance to new heights thanks to their strong multi-modal understanding and instruction-follownig, making them much more generalized in REC tasks. Many recent VLMs now build visual grounding directly as an important training objectives: Kosmos-2 (39), Qwen-VL (5), and Gemini (15; 22) are trained with bounding boxes annotations, whereas MoLMo (10) and RoboPoint (54) specialise in point-based localization. While REC datasets (23; 52; 33; 44; 30; 40) serve as a useful reference for evaluating the localization capabilities of VLMs, they primarily focus on basic object-level grounding in everyday scenes.

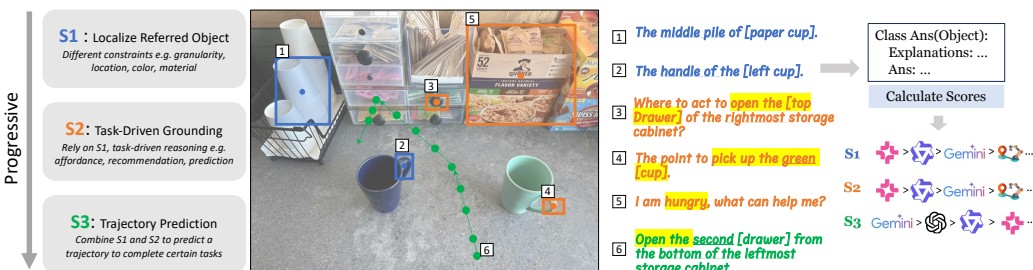

Figure 2: **A Hierarchical Framework for Visual Grounding in Embodied Reasoning.** We propose a three-stage progression: **S1** (object localization) localize objects explicited referred to in the text, with some conditions like granularity and appearance; **S2** (task-driven grounding) builds on S1 to infer locations used in specific task, which may not be explicitly referred to in the text ; and **S3** (visual trace prediction) combines S1 and S2 to generate executable motion plans. Underlined text denotes the *referred object* that needs to be localized (S1), while  yellow highlights  indicate task-contexts in *task-oriented reasoning* (S2/S3).

Moreover, they lack coverage of embodied scenarios that require more nuanced forms of grounding critical for task-related understanding: e.g., task-driven localization, affordance grounding, and visual trace prediction.

### 2.2 Benchmarking Vision-Language Models for Embodied Reasoning

As Vision-Language Models (VLMs) are increasingly applied to embodied tasks, a growing number of benchmark studies have been introduced to evaluate their reasoning capabilities. However, as shown in Table 1, most existing benchmarks either rely on indirect evaluation formats such as multiple-choice questions (4; 12; 43), generate high-level language-based plans (48; 27), or reduce actions to predefined skill sets (48). Localization in robotic scenarios has been explored by (32; 54), but these efforts are limited to indoor environments and focus only on simple object localization or vacant space detection, where we will show is not enough as visual grounding for *embodied tasks* (Section 3). The most recent benchmark is RefSpatial (58), but it focuses only on spatial relations, including localizing objects and placement. In our work, we aim to construct a hierarchical benchmark to evaluate critical visual grounding abilities essential for embodied reasoning, which provides rich and meaningful signal for the ability of current models.

## 3 Hierarchical Definition of Visual Grounding for Embodied Reasoning

In this section, we present a three-stage hierarchical framework that captures essential visual grounding capabilities for embodied reasoning. The stages are arranged in increasing complexity, with each level building upon previous ones. For example, the hierarchy derived from a household-robot task is shown in Figure 2, with additional examples from other domains illustrated in Figure 3. For each stage, we (i) define the specific visual grounding abilities it encompasses, (ii) provide relevant subclasses and scenario-specific examples, and (iii) highlight its importance by identifying existing embodied policy approaches that depend on these capabilities, either directly or indirectly.

### 3.1 S1: Referred Object Localization

Stage S1 focuses on identifying and localizing the **specific objects** in a scene as referenced by the language instruction. S1 aligns closely with referring expression comprehension (REC) (47) tasks commonly studied in the literature (47; 23; 32). In practice, language often includes additional constraints to disambiguate the target object, such as spatial cues, color, or material properties. Moreover, references may vary in granularity, ranging from whole objects to object parts (44). For example, in the household task shown in Figure 2, "the middle pile of paper cups" includes a location-based constraint, while "the handle of the left cup" involves both part-level and spatial restrictions.

We further divide S1 into three main categories: object without ambiguity (single object), object with constraints, and object part (See Figure 4 for examples of different subclasses).

S1 represents the most fundamental visual grounding capability required for embodied tasks: mapping language references to visual entities. It is an essential skill for nearly all open-vocabulary, **language-guided policies**, e.g. RT-series (7; 6; 36), OpenVLA (25),VIMA (21) and SayCan (2). As these models must first localize the referred object implicitly or explicitly before reasoning about or interacting with it.

### 3.2 S2: TASKED-DRIVEN GROUNDING

S2 goes beyond the explicit reference grounding in S1 by moving to a **task-driven visual grounding**: determining which object or part of an object is relevant for the task and pinpointing where to interact with it. Unlike S1, the entity to be localized may not be explicitly mentioned in the instruction, so it needs **reasoning** over target object and understanding of the action **affordance**.

S2 challenges the model to recognize action-relevant locations such as handles, buttons, or lids—based on contextual cues, even when these are not directly referred to in the instruction. For example, the command "open the top drawer" requires the model to (1) identify which drawer is being referred to (as in S1), and (2) localize the appropriate part to interact with (e.g., the handle). In another example, when given "I'm hungry, help me," the model must infer that a visible food item should be retrieved and localize where to grasp it. Thus, the essence of S2 lies in perceiving the affordances of objects and leveraging the **task context** to ground where to act. We further divide S2 into three categories: affordance, contact, and recommendation/safety grounding (See Figure 4).

Stage S2 highlights how embodied visual grounding differs from standard computer-vision grounding. A model that excels here shows a basic sense of how to interact with the physical world, links its perception to the task, and reasons simply about where to act. S2 itself captures visual affordance understanding, a key ingredient for general-purpose manipulation (11; 34; 18). Beyond that, it underpins many modern, versatile robot policies, e.g. VLAs (25; 36): even when the policy is not asked to output an affordance map, the robot still must know the right spot to act on.

### 3.3 S3: TASK-DRIVEN VISUAL TRACE PREDICTION

Building on the capabilities developed in S1 and S2, stage S3 assesses if VLM can plan accordingly to complete the instructed tasks. Given a task, the model must produce a **coarse 2D visual trace** that outlines how the task should be completed. Extending beyond S2, S3 introduces a temporal component: the agent must integrate object understanding, affordance reasoning, and prior decisions into a good motion path. While models proficient in S2 may handle simple pick-and-place tasks, S3 demands a more complete understanding of how to act e.g. generating a visual trace to wipe a table with a sponge or open and close a drawer requires movement beyond a fixed action spot.

Visual trace is an important intermediate policy disentangled from low-level motor actions. In particular, 2D visual traces have emerged as an increasingly valuable form of high-level intermediate representation. For example, RT-Trajectory (16) and Robotic Visual Instruction (29) use 2D trajectories as human-interpretable instructions to guide robots in task execution. Similarly, works such as ATM (46), Hamster (28), TraceVLA (57), Im2Flow2Act, Motion-before-Action (42), General-Flow (53) and Molmo-Act (26) incorporate 2D visual trace prediction as a critical stage, enhancing policy interpretability and enabling broader generalization compared to purely language-based or implicit state representations. Robobrain (20) includes predicting visual traces as an important fine-tuning stage to train planners in robotics tasks.

## 4 BENCHMARK CURATION, CANDIDATE MODELS AND EVALUATION METRICS

Our benchmark is made up of datapoints like $(\texttt{S, subclass}, \langle\texttt{Img}\rangle, \langle\texttt{question}\rangle, \langle\texttt{mask}\rangle)$, where $\texttt{S} \in \{\texttt{S}_1, \texttt{S}_2, \texttt{S}_3\}$. $\texttt{subclass}$ defines which specific subclass in the stage this datapoint belongs to. The left three attributes represent the input image, the question (description of the task), and the ground-truth polygon-based segmentation mask for the question.

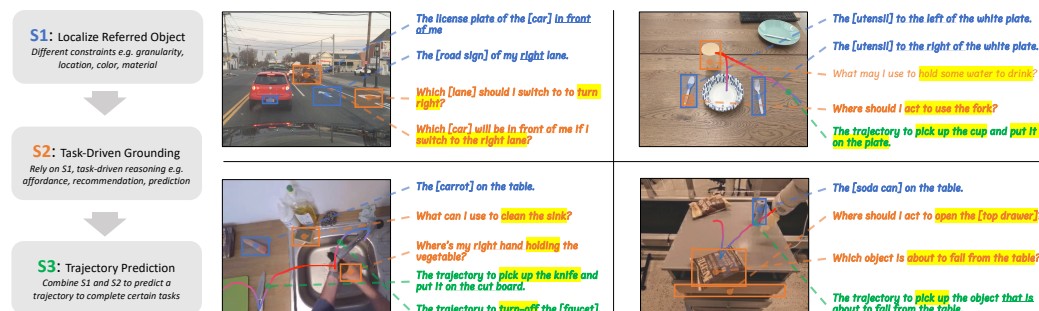

Figure 3: **More Examples for Three-Stages Grounding across Embodied Tasks**: We illustrate more examples across driving, kitchen, and robotic domains that align with our three-stage hierarchy.

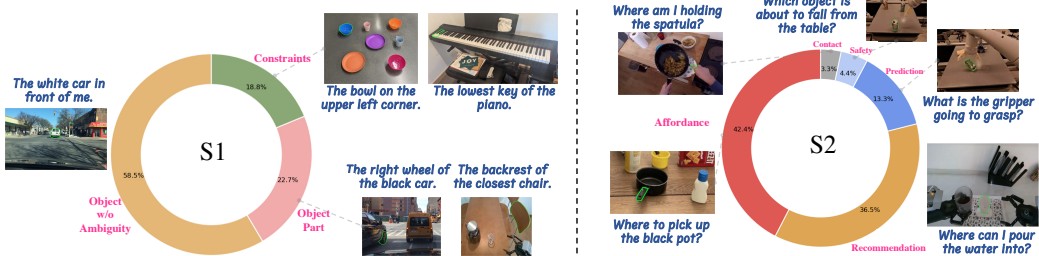

Figure 4: **Examples and Distributions of S1 and S2 Subclasses**: here we show examples of subclasess for S1 (object w/o ambiguity, object part, and object with constraints in e.g. locations, color) and S2 (affordance, prediction, safety, contact and recommendation); and also the % of them in the each stage.

For S1 and S2, we collect 501 question–answer (QA) pairs across five diverse datasets (around 230 for S1 and 270 for S2): Where2Place (54) (Apache 2.0), Ego4D–EPIC-Kitchens (9) (CC BY-NC 4.0), BDD100K (51) (CC BY-NC), AgiBot (8) (CC BY-NC-SA 4.0), and RT-1 (7) (CC BY-NC-SA 4.0). Each dataset contributes around 50 images and corresponding QA pairs, covering domains such as indoor scenes, kitchen manipulation, autonomous driving, and robotic control. This multi-domain composition ensures diversity in both visual context and embodied reasoning challenges.

We construct our benchmark around four key embodied tasks, using data from the following datasets: robot manipulation 🤖 from RT-1 (7), DROID (24), and AgiBot (8); household environments 🏠 from Where2Place (54); kitchen activities 🍳 from EPIC-Kitchens (9); and driving scenes 🚗 from BDD-100K (51). From these datasets, we extract image frames and select high-quality sample (e.g. filtering out those with motion blur or unclear visuals) o build our benchmark.

**Annotation:** For the first two stages, each datapoint is manually annotated using standard polygon-based segmentation tools (14). Guided by example prompts and a predefined set of stages and subclasses for each dataset, human annotators generate a natural language question, assign the appropriate stage and subclass, and provide an accurate polygon-based mask as the ground-truth answer. To reduce potential bias in language descriptions, we use GPT-4o (1) to rewrite prompts in a clearer and more formal manner, helping to minimize ambiguity and errors.

For S1 and S2, we collect 501 question–answer (QA) pairs across five diverse datasets (around 230 for S1 and 270 for S2): Where2Place, Ego4D-EpicKitchen, BDD100K, AgiBot, and RT-1. For S3, we extract frames on the AgiBot, DROID, and RT-1 datasets, collecting 100 questions targeting visual trace prediction tasks for robotic arms in images. In addition to the question and the predicted visual trace, we annotate the starting 2D position of the robot arm to guide the model with the initial configuration. We put more details and more examples about the dataset in the appendix.

**Candidates Models** The candidate models can be categorized by their output format. While most general instruction-following vision-language models are capable of both point and bounding box predictions, they tend to perform significantly better on one format over the other according to (54).

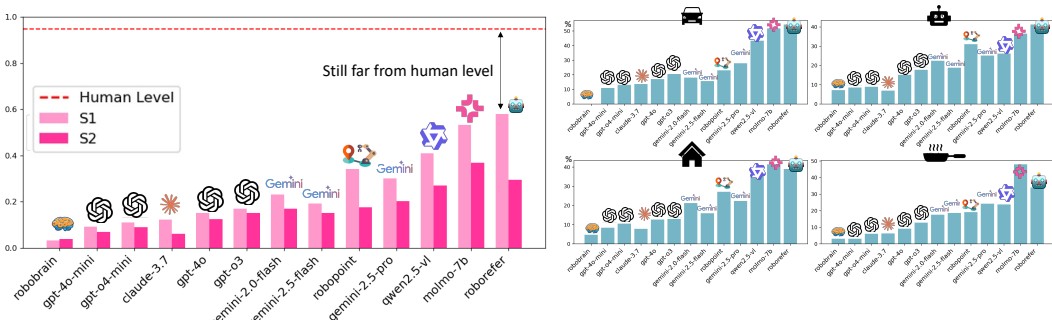

Figure 5: **Performance on S1 and S2 for Different VLMs**. **(Left)** Model scores on S1 and S2 tasks. RoboRefer-SFT-8B, MoLMO-7B, Gemini-2.5-Pro, and Qwen-2.5-VL significantly outperform other models. **(Right)** Average scores across S1 and S2 of different model in four distinct scenarios.

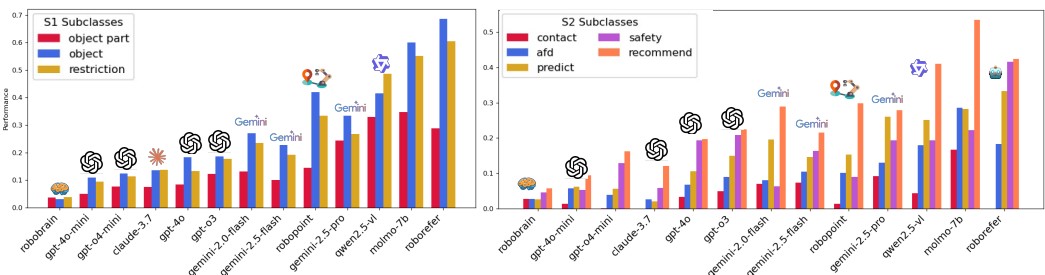

Figure 6: **Per-subclass performance:** (Left) In **S1**, every model shows a clear drop in accuracy when localizing *object parts*. (Right) In **S2**, the *contact* and *affordance* (afd) subclasses remain the most challenging, whereas *recommendation* is relatively easier for most models because it relies on simple language-level reasoning. MoLMO (10) and Qwen-VL (5) attains the highest score on most subclasses.

For instance, MoLMO (10) and RoboPoint (54) excel at point prediction but struggle with bounding box outputs, whereas GPT-4o (1) more easily produces bounding boxes than points.

(VLMs that Output BoundingBoxes): We have strong black-box models: GPT-4o/4o-mini (1), GPT-o3/o4-mini (37) (API version), Claude-3.7-Sonnet (3), Gemini-2.0-Flash (15), Gemini-2.5-Flash/Pro (22); and also strong close sourced model like Qwen-2.5-VL-32B-Instruct (5). For these models we prompt them to output **only one** bounding box that best match the location described in the language descrption.

(VLMs that Output Points): Recent models try to do fine-tuning based on point-based grounding due to its scalability (10). We include Molmo-7B-D (10) as a general-purpose-pointing model, and RoboPoint (54), which is carefully fine-tuned using robotic data. For these models, we prompt them to output one or a few points that best correspond to the target described in the language input. All prompt templates used for evaluation are provided in the Appendix C.

**Evaluation Metrics** To ensure fair comparison across different output formats (points vs. bounding boxes), we propose a normalized IoU metric for S1 and S2 that accounts for the difficulty of covering irregular masks with rectangular boxes. This improves upon the biased point-in-mask evaluation used in prior work (54). For S3, we evaluate visual trace quality using two methods: human ratings on a 1–5 scale and GPT-o4-mini based assessments guided by predefined criteria (e.g. the overall direction of the trajectory, the keypoint coverage, and the task feasibility). Further implementation details can be found in the Appendix D.

## 5 EXPERIMENTS

In this section, we present several key insights derived from quantitative evaluations and visualizations based on testing different VLMs on the PIO benchmark. We organize this section into blocks of key findings, each followed by supporting evidence and analysis.

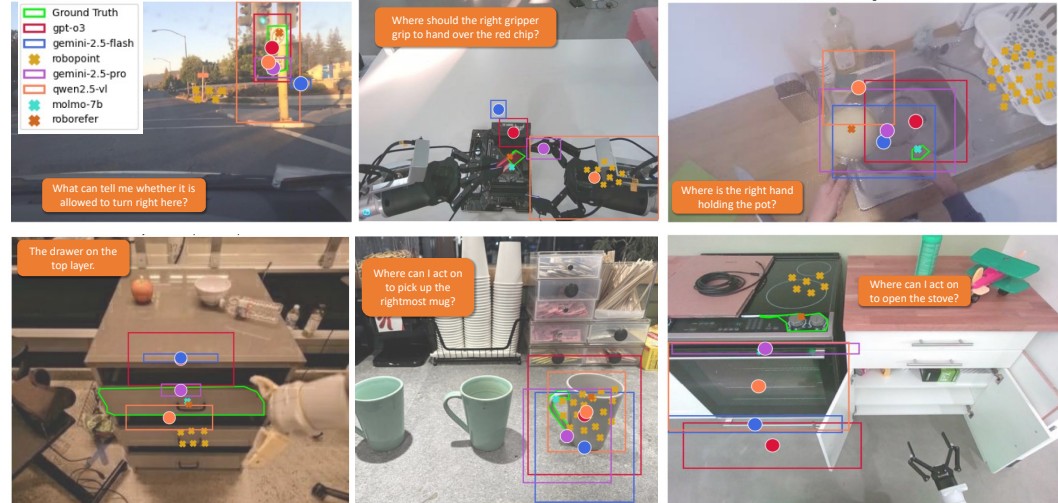

Figure 7: **Example predictions from different models on S1 and S2 tasks:** here we show some visualization of predicted bounding boxes and points of different VLMs for S1 and S2. More visualizations of more models are put in the Appendix 9 and 10 .

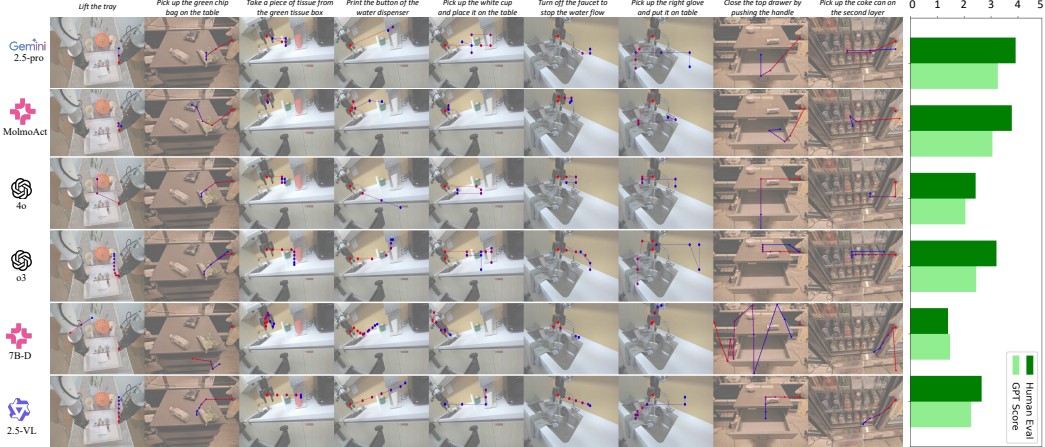

Figure 8: **Visualizations and Scores on S3**: We present visualizations of S3 visual trace predictions from different models. The scores are shown on the right. Gemini-2.5-Pro, MolmoAct and GPT-o3 outperform other models, while MoLMO and Qwen, despite their strong performance on S1 and S2, struggle with temporal visual trace prediction compared to stronger general-purpose models.

> **Finding 1**: For S1 and S2, models that incorporate explicit grounding supervision such as RoboRefer, MoLMO-7B-D, Gemini-2.5-Pro, and Qwen-2.5-VL consistently achieve the highest, outperforming more general-purpose VLMs such as GPT-4o and Claude-3.7. This underscores the importance of grounding data when precise spatial reasoning is required .

Figure 5 illustrates the overall performance of all candidate models on S1 (Referred Object Localization) and S2 (Task-Driven Localization). RoboRefer, MoLMO and Qwen consistently outperform other models across both tasks. Within the GPT series, GPT-o3 achieves the best results, likely due to its enhanced reasoning capabilities, whereas smaller variants such as GPT-o4 mini and GPT-4o mini perform noticeably worse. In the Gemini family, Gemini-2.5 Pro significantly outperforms both Gemini-2.0 Flash and Gemini-2.5 Flash, and also slightly surpasses RoboPoint.

Although GPT-4o and Claude-3.7 have demonstrated strong performance in language-based embodied reasoning on prior benchmarks (27; 48), they fall short on `PIO` where previse visual grounding is needed.

> **Finding 2**: (i) Across all subclasses, every model exhibits a clear performance drop from S1 to S2. (ii) Two critical embodied skills suffer most: in S1 they often miss the correct *object part* to point to, and in S2 they struggle with *affordance* and *contact* prediction.

Looking at the performance of different subclasses in S1 and S2 (Figure 6), we find that most models do not show much performance drop on tasks that only require simple language reasoning—such as "object with restriction" in S1 and "recommendation" in S2. This is likely because the models can handle basic reasoning before predicting the bounding box or point.

However, when it comes to more detailed tasks like localizing object parts, most models perform significantly worse (left side of Figure 6). Even strong models like Qwen (5) and MoLMO (10) score below 0.5. In S2, while most models handle recommendation well, they struggle with affordance and contact prediction. Although MoLMO is the best in affordance, it still scores below 0.4.

These findings suggest that more attention should be paid to improving the grounding abilities of models, especially for fine-grained tasks such as object part localization, affordance, and contact. These are crucial skills if vision-language models are to truly act as grounded agents capable of interacting with the real world.

> **Finding 3**: S3 requires models to integrate single-target grounding into coherent visual trace generation. While S1 and S2 are *necessary* prerequisites, they are not *sufficient* for a model to succeed in S3. Gemini-2.5-pro shows promising results in S3 and also performs well in S1 and S2. In contrast, MoLMO and Qwen, the top-performing models in S1 and S2, fail in S3.

To evaluate the performance in S3 (visual trace prediction), we visualize model outputs together with human ratings and GPT-based assessments in Figure 8. Unlike S1 and S2, where fine-tuned models such as MoLMO and Qwen show strong grounding capabilities, these models underperform in visual trace generation. This suggests that although they can localize specific targets effectively, they struggle to extend this capability into multi-step, temporally coherent planning.

In contrast, we find that Gemini-2.5-Pro (22) and GPT-o3 (26) (slightly outperforming GPT-4o) generate more reasonable trajectories that (1) follow the correct overall direction and (2) successfully reach the target objects. Gemini-2.5-Pro achieves almost 4 out of 5 in Figure 8 (right), it can be attributed to the inclusion of embodied data and grounding data in the strong model (43). This indicates that strongly pre-trained VLMs such as GPT-based models excel at handling complex tasks that involve both grounding and planning, even without fine-tuning over 2D trajectory data. In comparison, smaller models like MoLMO-7B and Qwen, while effective at isolated grounding tasks, struggle to jointly perform grounding and visual trace planning, limiting their utility in more integrated embodied scenarios (28; 16; 26).

> **Guideline for Model Users**: For task does not require action generation (e.g. pick and place based on points), prioritize models that perform well on S1 and S2. If the goal is to train a robot policy using a VLM backbone, models that perform well on S3 - even if weaker on S1 / S2, could be better candidates. Furthermore, users might consider adding grounding data and fine-tuning to further improve performance for different stages.

## 6 CONCLUSIONS

In this work, we introduce `PIO`, a new benchmark designed to evaluate vision-language models (VLMs) in precise visual grounding and embodied reasoning tasks. By decomposing the evaluation into three stages, referred object localization (S1), task-driven localization (S2), and visual trace prediction (S3), our benchmark reveals fine-grained insights into embodied reasoning ability of different VLMs in regard of precise grounding. Through extensive quantitative and qualitative evaluations of over ten state-of-the-art VLMs, we uncover several key findings: models fine-tuned with grounding supervision excel at S1 and S2, but struggle with S3; in contrast, strong generalist models perform better in multi-step reasoning and planning tasks, despite weaker spatial precision.

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

# A DETAILS OF BENCHMARK CURATION

We collect frames from the following datasets:

- Driving
    - **BDD-100k** (51): front camera view of driving scenes, CC BY-NC
- Household
    - **Where2Place** (54): object placement in indoor scenes, Apache 2.0
- Kitchen
    - **EPIC-Kitchens** (9): egocentric kitchen interactions, CC BY-NC 4.0
- Robot Manipulation
    - **RT-1** (7): real-world robot arm manipulation, CC BY-NC-SA 4.0
    - **DROID** (24): real-world robot arm manipulation, MIT License
    - **AgiBot** (8): real-world humanoid robot manipulation, CC BY-NC-SA 4.0

## A.1 DATA COLLECTION FOR S1 AND S2

We use RoboFlow (14) to collect human-annotated masks in polygonal format, using images from BDD100K, Where2Place, EPIC-Kitchens, RT-1 and AgiBot. The initial prompts for S1 and S2 are generated by human annotators following a structured hierarchy and guided by subclass examples. Table 2 summarizes the number of annotated images and question-answer (QA) pairs across different datasets. Also, Table 3 shows examples of questions of different stages and sub-classes.

| Dataset | #Images | #QAs |
|---|---|---|
| Where2Place | 50 | 157 |
| Epic-Kitchen | 50 | 108 |
| BDD-10k | 51 | 135 |
| AgiBot | 36 | 45 |
| RT-1 | 54 | 56 |
| **Total** | **241** | **501** |

Table 2: Number of annotated images and QA pairs for each dataset used in S1 and S2.

## A.2 DATA COLLECTION FOR S3

For S3, we concentrate on robot manipulation, as it is the most suitable domain for visual trace generation. We collect roughly 50 tasks from three datasets: AgiBot (8), DROID (24), and RT-1 (7). Each task is posed to the models evaluated as a question paired with an image; examples are given in Table 4. We did not include ground-truth trajectories because robot-manipulation problems are inherently multimodal: A single objective can be achieved through many valid trajectories. Providing a canonical answer would be potentially misleading.

# B TESTED MODELS

We show all candidate models used in this paper in Table 5. For open-sourced models we ran them locally on one Nvidia-A6000 GPU, for close-sourced models we use the provided API.

Table 5: Models evaluated in this study.

| Model Name | Model ID | Size | Link |
|---|---|---|---|
| GPT-4o | gpt-4o | Unknown | [link] |
| GPT-4o-mini | gpt-4o-mini | Unknown | [link] |
| OpenAI o3 | o3 | Unknown | [link] |
| OpenAI o4-mini | o4-mini | Unknown | [link] |
| Claude-3.7-Sonnet | claude-3-7-sonnet | Unknown | [link] |
| Gemini 2.0 Flash | gemini-2.0-flash | Unknown | [link] |
| Gemini 2.5 Flash | gemini-2.5-flash-preview-05-20 | Unknown | [link] |
| Gemini 2.5 Pro | gemini-2.5-pro-preview-05-06 | Unknown | [link] |
| Molmo-7B-D | allenai/Molmo-7B-D-0924 | 7B | [link] |
| RoboPoint | wentao-yuan/robopoint-v1-vicuna-v1.5-13b | 13B | [link] |
| Qwen 2.5 VL-32B Instruct | Qwen/Qwen2.5-VL-32B-Instruct | 32B | [link] |
| Robobrain | BAAI/RoboBrain | 7B | [link] |
| RoboRefer | RoboRefer-8B-SFT | 8B | [link] |
| MolmoAct | allenai/MolmoAct | 7B | [link] |

## C PROMPTS

Here we provide all the prompts used in this paper including:

- S1, S2 prompts for different models
- S3 prompts for selected models
- S3 auto evaluation prompts

**S1 S2 Prompt - GPT**

```
You are an agent skilled in spatial reasoning and object localization.
Given an image and a text description, output **only** the bounding box
    that best matches the description, formatted as a tuple of
    normalized integers: `(min_x, min_y, max_x, max_y)`.

- Coordinates follow OpenCV format: `x` = column index, `y` = row index.
- **All values must be normalized to [0, 1]**.
- **Do not include any additional text or explanation.**

The output should be in the format of:

class ReturnedCoordinate(BaseModel):
    explanations: str
    min_x: float
    min_y: float
    max_x: float
    max_y: float

> Image and description are provided below.
> Language description:
{question}
> Answer:
```

**S1 S2 Prompt - Qwen**

```
Locate to the bounding box of the area / object described in the
    following question in json format: {question}.

 - Do not output "there are none", the object always exists!
```

```
 - Please only output one bounding box !
```

### S1 S2 Prompt - MoLMO

```
Point to one or more points that best correspond to the following
    question:
{question}
Note: The object referenced in the question always exists, so do not
    respond with "I don't know." / "There are none"

Please DO NOT output any other texts besides points!
```

### S1 S2 Prompt - RoboPoint

```
Please locate several points better fit the following question: {
    question},
Your answer should be formatted as a list of tuples, i.e. [(x1, y1), (x2
    , y2), ...], where each tuple contains the x and y coordinates of a
    point satisfying the conditions above.
The coordinates should be between 0 and 1, indicating the normalized
    pixel locations of the points in the image.
```

### S3 Prompt - GPT

```
You are an agent to help me generate rough 2D visual trace to guided the
     robot to compelte the tasks. Given a observation image, you are
    told the current 2D location of the end-effector (marked in red in
    the image).
You need to output a sequence of 2D points starting from the current
    position as the rough trajectory to complete the task.

# Return Format
class ReturnedTrajectory():
    explanations: str  # analyzing process, e.g. decompose the task
    trajectory: [(float, float)]  # normalized points starting from red
        marker

# Some Tips:
(1) take care of the contact point when interacting with the object, e.g
    . handle of cup and bottle
(2) the trajectory should faithfully reflect the scale on the 2D pixel
    space
(3) you can first detect the important point and put the reasoning
    process in the explanations

# Task Description
{task}

# Current End-effector 2D position (marked in red in the image)
{eepose}

# You Answer as 2D Trajectory to Complete the Task
```

### S3 Prompt - Qwen

```
Given the task description and the image observation, you need to output
     a sequence of 2D points that can complete the task.
```

```
You will also be offered the starting points of the current end-effector
    (annotated in red marker in the given image) and its accurate 2D
    position.
You need to return 5~10 points (x,y) as trajectory starting from the
    current position. Do not return anything but points!

 - Do not output "there are none", the trajectory always exists!

## Task Description:
{task}

## Current End-effector 2D position (marked in red in the image):
{eepose}

Please return your answer here, the format should be:
{'explanation':'...', 'trajectory':[(x1, y1), (x2, y2), ...]}

# Answer:
```

**S3 Prompt - MoLMO**

```
Based on the input image, points to a sequence of 2D points in format of
    html points as a rough trajectory to complete the following task: {
    task},
the starting point is annotated in red in the image, and the position is
    {eepose}.
Your output trajectory should start from it.

Your answer is:
```

**S3 AutoScore Prompt**

```
### Role
You are the evaluator responsible for scoring a model-generated
    trajectory.

### Input
- **Image**: Shows the current scene.
  - The task description is displayed at the top of the image.
  - The trajectory originates at the red point (robot gripper) and
     gradually shifts to blue as it progresses.

### Output
Provide:
1. **Score** - an integer from **1 to 5**.
2. **Rationale** - a brief explanation (2-3 sentences) justifying the
    score.

| Score | Interpretation |
|-------|----------------|
| 1 | Poor trajectory - nonsensical path that cannot accomplish the task
    . |
| 2 | Direction shows faint promise, but overall execution is
    unsatisfactory. |
| 3 | General direction and keypoints are reasonable, yet important
    flaws remain. |
| 4 | Largely correct; minor inaccuracies at some keypoints. |
| 5 | Excellent - visits all required keypoints and should successfully
    complete the task. |
```

```
### Evaluation Criteria
- **Directional accuracy**
- **Keypoint coverage**
- **Task feasibility**

*(Keep explanations concise and focused on these criteria.)*

### Return Format
class Ans(BaseModel):
    explanation: str  # include your explanations
    score: int        # a score from 1 to 5

### Your Answer
```

## D  EVALUATION METRICS

**S1 and S2**: It is straight-forward to compare VLMs that output the same format (e.g. use IoU for boundingboxes), it need more careful design to handle cases where models have different formats of outputs. In (54), the authors evaluate performance using the percentage of predicted points that fall within annotated masks. For bounding boxes, they uniformly sample points within the box to enable comparison. Although it may not affect the final ranking in (54), the metric itself is flawed: it introduces a bias toward point-based methods, the irregular shape of the mask makes it challenging for bounding boxes to achieve high scores, as they cannot fully cover the masked area. In our experiments for S1 and S2, we propose a **normalized IoU** metric to replace the evaluation method used in (54). Specifically, we normalize the IoU score between the predicted bounding box and the ground-truth mask using a reference term: the IoU between the tightest bounding box enclosing the ground-truth mask and the mask itself. The final score is defined as: $s = \frac{\text{IoU}(\text{bbx}_{\text{pred}}, \text{mask})}{\text{IoU}(\text{bbx}_{\text{tight}}, \text{mask})}$. This normalization accounts for the inherent difficulty of tightly covering irregular-shaped masks with rectangular boxes, allowing for a fairer comparison.

**S3**: To evaluate the quality of the proposed trajectories generated in **S3**, we employ two evaluation metrics. First, human annotators are asked to score the outputs of various VLMs on a scale from 1 to 5. Second, we leverage GPT-o4-mini to assess and rate the trajectories based on prompts guided by predefined evaluation criteria. The prompts we used are in Section C.

## E  MORE EXAMPLES OF S1 S2 PREDICTIONS

We show more examples in Figure 9 for underperformed models like GPT-4o, GPT-4o-mini, Claude-3.7, Gemini-2.0-flash, GPT-4o and in Figure 10 for top rated models e.g. Qwen, Molmo and Gemini-2.5-pro.

## F  MORE EXAMPLES OF S3 RESULTS

In Figure 11, we show more model prediction visualizations of trajectory prediction in S3.

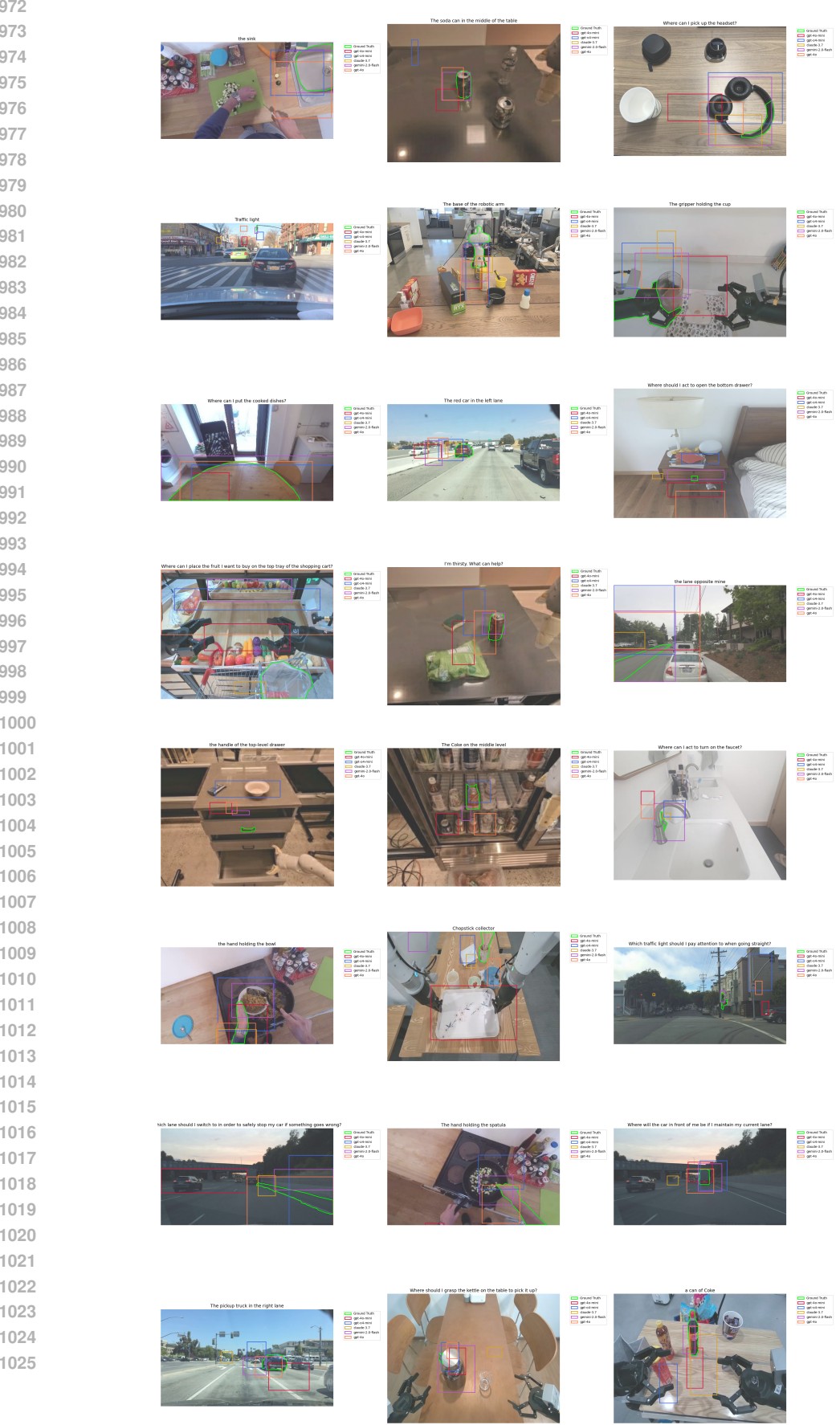

19

Figure 9: Model prediction visualization for GPT-4o, GPT-4o-mini, Claude-3.7, Gemini-2.0-flash, GPT-4o, which underperform other models.

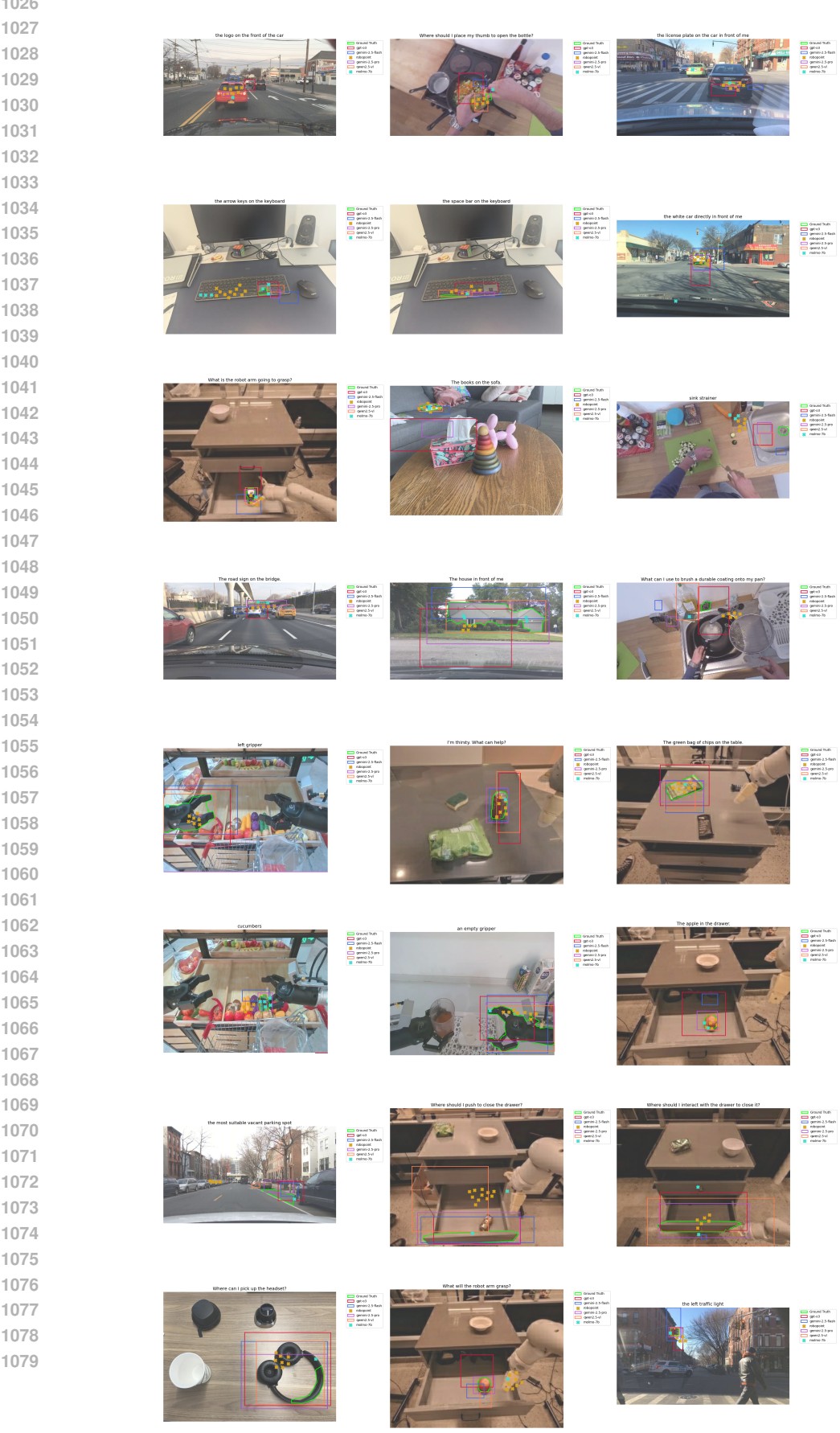

Figure 10: Model prediction visualization for top rated models, e.g. Qwen, Molmo and Gemini-2.5-pro.

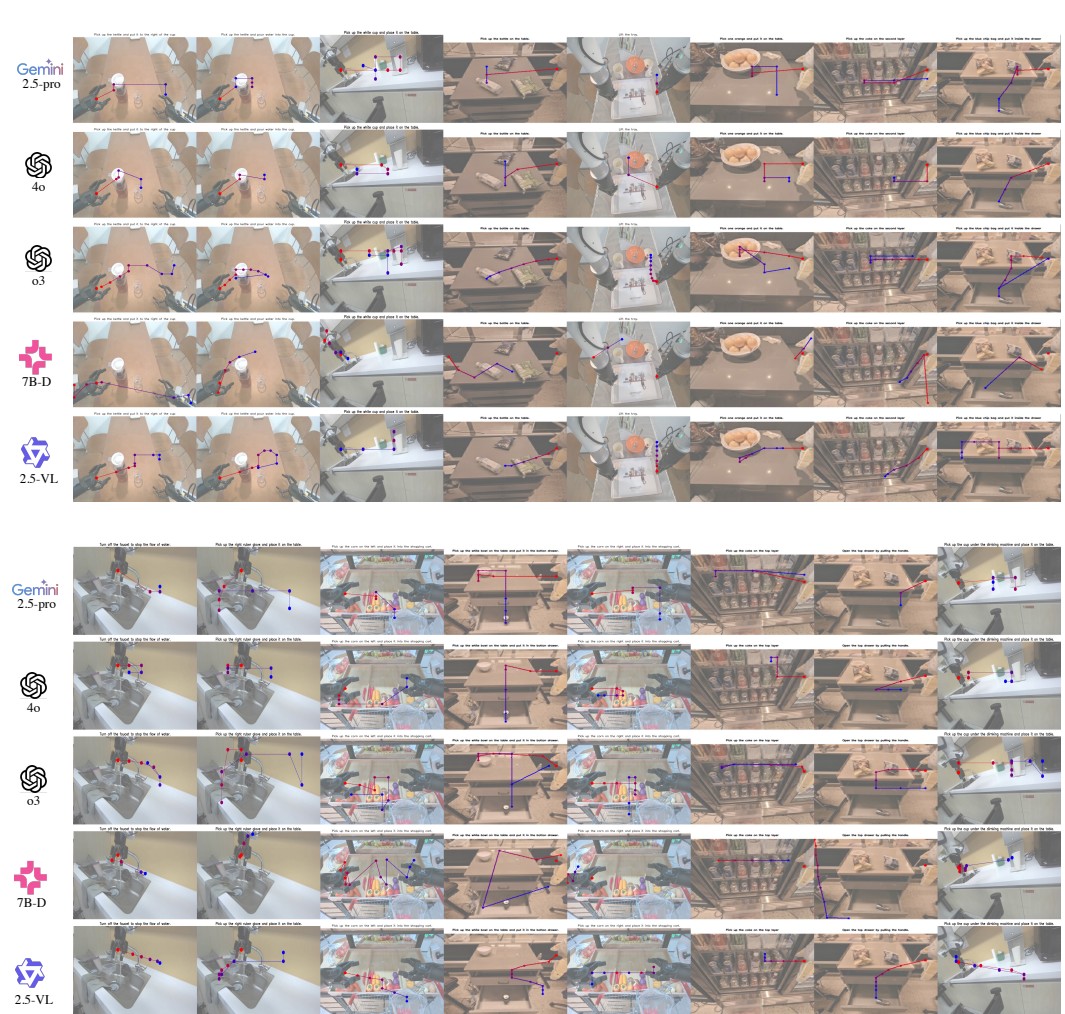

Figure 11: More model prediction visualizations of S3

| Stage | Subclass | Datasource | Prompt |
|---|---|---|---|
| S1 | object | epickitchen | The hand holding the spatula |
| S1 | object | agibot | The empty space in the center of the wooden tray. |
| S1 | object | bdd10k | The stop sign in front of me |
| S1 | object | where2place | an empty pot |
| S1 | object | rt1 | The green bag of chips on the table. |
| S1 | object | where2place | the toothpaste |
| S1 | object | agibot | The gripper holding the cup |
| S1 | object | bdd10k | the lane to my right |
| S1 | object | epickitchen | the sink |
| S1 | object | where2place | a pair of scissors |
| S1 | object | bdd10k | Car covered with multiple text inscriptions |
| S1 | object | bdd10k | the black pickup truck to my right |
| S1 | object | epickitchen | The only utensil on the table |
| S1 | object | bdd10k | the nearest black trash bag |
| S1 | object part | bdd10k | the left turn signal light of the car in front of me |
| S1 | object part | rt1 | The handle of the second-level drawer |
| S1 | object part | epickitchen | hand holding the spatula |
| S1 | object part | epickitchen | sink strainer |
| S1 | object part | where2place | The tail of the pink balloon dog. |
| S1 | object part | where2place | The bottom left handle of the controller. |
| S1 | restriction | agibot | The candies on the plate. |
| S1 | restriction | where2place | the first flight of the staircase |
| S1 | restriction | bdd10k | the car straddling the lanes |
| S1 | restriction | where2place | the non-empty mug |
| S1 | restriction | bdd10k | the black car turning left at the crossroads |
| S1 | restriction | where2place | the pillow at the leftmost position on the sofa |
| S2 | afd | agibot | Where should I press to flush the toilet? |
| S2 | afd | agibot | Where can I place the fork after dinner? |
| S2 | afd | where2place | Where should you act to open the second-highest drawer? |
| S2 | afd | bdd10k | The vacant space between the two cars in the left lane |
| S2 | afd | where2place | Where should I act to pick up the portafilter? |
| S2 | afd | rt1 | Where should I push to close the drawer? |
| S2 | afd | where2place | Where should I interact with the fork in order to pick it up? |
| S2 | afd | where2place | Where should I act to open the bottom drawer? |
| S2 | afd | bdd10k | Switch into the left-turn lane. |
| S2 | afd | where2place | Where should I interact with the mug to open its cap? |
| S2 | predict | bdd10k | Where will the car ahead of me be if I switch one lane to the left? |
| S2 | predict | rt1 | What is the robot arm going to grasp? |
| S2 | predict | bdd10k | Which car will be in front of me if I switch to the left lane? |
| S2 | predict | rt1 | What is the robot arm going to grasp? |
| S2 | recommend | agibot | What is the best gripper to use for picking up a carrot? |
| S2 | recommend | where2place | What can I use to hold water for drinking? |
| S2 | recommend | epickitchen | Where should I place my new pan for cooking? |
| S2 | recommend | where2place | My hand is wet. What can I use to dry it? |
| S2 | recommend | where2place | I'm feeling sleepy but I need to keep working. What can help me feel more energetic? |
| S2 | recommend | bdd10k | Which lane should I switch to if I follow the black car? |
| S2 | recommend | where2place | What can I use to hold some water? |
| S2 | contact | agibot | The point where the right gripper grasps the tray? |
| S2 | safety | bdd10k | What should I check before making a U-turn here? |
| S2 | safety | bdd10k | If something goes wrong with my car, where should I park it to wait for help? |

Table 3: Example 50 out of 500 prompts in S1 and S2

| Datasource | Image ID | Prompts |
|---|---|---|
| agibot | 0 | Pick up the kettle and pour water into the cup. |
| | | Pick up the kettle and put it to the right of the cup. |
| | | Pick up the cup. |
| agibot | 1 | Pick up the corn on the left and place it into the shopping cart. |
| | | Pick up the corn on the right and place it into the shopping cart. |
| | | Move the trolley to the right. |
| | | Move the trolley to the left. |
| agibot | 2 | Wipe the surface of the toilet lid with a cloth. |
| | | Wipe the tank lid of the toilet with a cloth. |
| agibot | 3 | Pick up the coke bottle and place it to the left of the ice red tea bottle. |
| | | Pick up the coke bottle and place it to the right space on the table. |
| agibot | 4 | Lift the tray. |
| agibot | 5 | Reach to the key on the left gripper. |
| | | Open the door. |
| droid | 0 | Press the button on the water dispenser to stop the water flow. |
| | | Pick up the white cup and place it on the table. |
| droid | 1 | Turn off the faucet to stop the flow of water. |
| | | Pick up the right rubber glove and place it on the table. |
| droid | 2 | Clean the table with the cloth. |
| droid | 3 | Pick up the cup under the drinking machine and place it on the table. |
| droid | 4 | Take a piece of tissue from the green tissue box. |
| | | Pick up the red cup and place it on the table. |
| rt1 | 0 | Pick up the bottle on the table. |
| | | Pick up the green chip bag on the table. |
| rt1 | 1 | Pick up the white bowl on the table and put it in the bottom drawer. |
| rt1 | 2 | Close the top drawer by pushing the handle. |
| rt1 | 3 | Pick up the green soda can on the table and put it to the left of the silver can. |
| rt1 | 4 | Pick up the apple and put it to the left of the water bottle. |
| | | Pick up the apple and put it to the right of the water bottle. |
| | | Pick up the water bottle and put it to the right of the apple. |
| | | Pick up the water bottle and put it to the left of the apple. |
| rt1 | 5 | Pick up the soda can in the drawer and put it on the vacant space on the top half of the table. |
| | | Pick up the blue chip bag and put it inside the drawer. |
| | | Open the top drawer by pulling the handle. |
| | | Close the bottom drawer by pushing the handle. |
| rt1 | 9 | Pick one orange and put it on the table. |
| | | I am hungry, please put something on the table for me to eat. |
| rt1 | 10 | Clean the table using existing object on the table. |
| rt1 | 11 | Pick up the coke on the top layer. |
| | | Pick up the coke on the second layer. |

Table 4: Examples of questions in S3: Visual trace prediction

