# OpenReview forum: "Point-It-Out: Benchmarking Embodied Reasoning for Vision Language Models in Multi-Stage  Visual Grounding"
_ICLR.cc/2026/Conference — Submitted to ICLR 2026_

### Official Review · Reviewer_dRu3 · 2025-10-30

**Soundness:** 4
**Presentation:** 4
**Contribution:** 4
**Rating:** 8
**Confidence:** 4

**Summary:**

The paper introduces Point-It-Out (PIO), a benchmark that evaluates embodied reasoning (ER) of VLMs via direct visual grounding rather than indirect multiple-choice or language-only formats. PIO defines a three-stage, hierarchical protocol: S1 (referred-object localization), S2 (task-driven pointing/ action affordance), and S3 (visual trace prediction / coarse 2D trajectories). The benchmark includes 600+ human-annotated datapoints that span indoor/household, kitchen, driving, and robotic manipulation settings and reports results for more than 10 models. Results show that models with explicit grounding supervision (MoLMO, Qwen-VL, RoboRefer) outperform general VLMs on S1/S2, while generalist reasoning models (Gemini-2.5-Pro, GPT-o3) perform better on the motion-oriented S3 task. The authors also propose a normalized IoU to fairly compare point vs. box predictions in S1/S2, and assess S3 performance through human ratings and a prompted LLM evaluator.

**Strengths:**

1. The core idea of evaluating embodied reasoning via explicit, precise visual grounding (points, bounding boxes, trajectories) as a direct output is a clear advancement over indirect, language-only, or multiple-choice evaluations.
2. The three-stage hierarchy is well-justified, capturing the natural progression of complexity from basic localization to affordance reasoning to temporal planning.
3. The extensive evaluation of 10+ VLMs and the derived insights regarding the trade-off between specialization (grounding-fine-tuned models excel at S1/S2) and generalization/planning (generalist models excel at S3) are valuable.
4. The paper is well-written, making contributions easy to understand. The figures are effective at conveying the core ideas and results.

**Weaknesses:**

1. While diverse, the size of the benchmark is relatively small compared to other recent embodied benchmarks (Table 1, RoboRefIt 10k, EmbSpatial-Bench 3.6k). Given the split across 4 scenarios and multiple subclasses, this small size may limit the statistical robustness of the findings across all fine-grained categories. A discussion on the challenges and potential plans for scaling the benchmark would be welcome.
2. The absence of GT trajectories is understandable, but the combination of human and LLM raters warrants inter-rater reliability reports.
3. The evaluation of visual traces in S3 relies on human and GPT-4o ratings. While justified by the multimodal nature of trajectories, the lack of an objective ground-truth path or a more rigorous metric is a limitation. Expanding on the limitation could strengthen the paper.

**Questions:**

1. Could more detail be provided on the human evaluation process for the S3 task? How many annotators rated each trajectory, and what was the inter-annotator agreement? Was the performance of the GPT-4o evaluator validated against the human scores to ensure its reliability?
2. Do higher S3 scores correlate with success rates in a lightweight simulator or a small real-robot eval? A correlation study could strengthen the causal claim.
3. The paper notes that S1/S2 proficiency doesn't translate to S3 success for models like MoLMO and Qwen. Could this also be related to their model architecture, or training data attribution?

---

> ### Author Response · Authors · 2025-11-21
> **Thanks for your careful review and valuable comments.**
>
> Thanks for you time to review our paper and your valuable comments, here we are happy to answer some questions you raised, and hopefully we will address some of your concerns:
>
> > While diverse, the size of the benchmark is relatively small compared to other recent embodied benchmarks (Table 1, RoboRefIt 10k, EmbSpatial-Bench 3.6k). Given the split across 4 scenarios and multiple subclasses, this small size may limit the statistical robustness of the findings across all fine-grained categories. A discussion on the challenges and potential plans for scaling the benchmark would be welcome.
>
> RoboRefIt is limited to around 10k examples mainly because it operates on video frames and does not cover as many diverse scenarios as our benchmark. EmbSpatial-Bench, while larger, is MCQ-based and can generate many questions from a single image, and it also relies heavily on simulated environments. In contrast, the current scale of PIO-Bench already provides a strong and consistent signal, as reflected in both qualitative visualizations and quantitative metrics.
>
> Further scaling PIO-Bench e.g., to support training stronger models, is entirely feasible. One practical direction is to leverage recent high-quality segmentation models such as SAM-2 to automate mask generation, followed by VLMs to assign labels.
>
> > The absence of GT trajectories is understandable, but the combination of human and LLM raters warrants inter-rater reliability
> reports.
>
> We put more details about the human survey in the Appendix (latest reversion, in red).
>
> > The evaluation of visual traces in S3 relies on human and GPT-4o ratings. While justified by the multimodal nature of trajectories, the lack of an objective ground-truth path or a more rigorous metric is a limitation. Expanding on the limitation could strengthen the paper.
>
> We agree. The multimodal nature of trajectories makes grounded auto-metrics difficult. We think the current evaluation shows reasonable signal regarding mode performance. We will think about further polish it with stronger VLM judgers with more detailed prompts.
>
> > Could more detail be provided on the human evaluation process for the S3 task? How many annotators rated each trajectory, and what was the inter-annotator agreement? Was the performance of the GPT-4o evaluator validated against the human scores to ensure its reliability?
>
>
> We put more details about the human survey in the Appendix (latest reversion, in red).
>
>
> > Do higher S3 scores correlate with success rates in a lightweight simulator or a small real-robot eval? A correlation study could strengthen the causal claim.
>
> We focus on coarse trajectories, and we do not claim that they directly translate into executable real-world actions. However, prior work such as Hamster [a], RT-Trajectory [c], and Molmo-Act [b] shows that predicted trajectories can guide robot behavior. Based on these findings, we assume that producing reasonable trajectories is correlated with higher likelihood of successful task execution, even if the mapping is not perfectly precise.
>
>
> > The paper notes that S1/S2 proficiency doesn't translate to S3 success for models like MoLMO and Qwen. Could this also be related to their model architecture, or training data attribution?
>
> Yes, we believe Molmo and Qwen underperform in s3 partly because they are not as strong as Gemini at handling cases that were unseen during training.
>
>
> [a] Hamster: Hierarchical action models for open-world robot manipulation
> [b] Molmoact: Action reasoning models that can reason in space
> [c] Rt-trajectory: Robotic task generalization via hindsight trajectory sketches

---

### Official Review · Reviewer_Gmue · 2025-10-31

**Soundness:** 3
**Presentation:** 2
**Contribution:** 2
**Rating:** 4
**Confidence:** 4

**Summary:**

The paper proposes Point-It-Out (PIO), a benchmark for evaluating the embodied reasoning abilities of vision-language models (VLMs) through precise visual grounding instead of multiple-choice tasks. Covering four domains—household, kitchen, driving, and robot manipulation—PIO defines a three-stage hierarchy: (S1) referred-object localization, (S2) task-driven grounding involving affordance or contact reasoning, and (S3) visual trace prediction for temporal reasoning. With over 600 human-annotated samples using polygon-based masks and fine-grained subclasses, the authors benchmark more than ten VLMs (e.g., GPT-4o, Claude-3.7, Gemini-2.5, MoLMO-7B, Qwen2.5-VL, RoboRefer) and find that models trained with explicit grounding supervision outperform general-purpose ones on spatial reasoning (S1/S2), while generalist models show advantages in temporal planning (S3).

**Strengths:**

The paper goes beyond MCQ-style or language-only benchmarks by enforcing pixel-level output (points, boxes, trajectories) that directly connects perception and action, addressing an underexplored gap in embodied reasoning evaluation.

The three-stage structure (S1–S3) is conceptually clear and mirrors increasing task complexity—from perception to reasoning to action planning—providing diagnostic insights into different reasoning stages.

The authors test a wide range of both closed and open VLMs, highlighting clear trends and limitations, and propose practical guidelines for model users.

**Weaknesses:**

The contributions are somewhat limited. Although the PIO benchmark addresses a gap in evaluating embodied reasoning through visual grounding, the concept itself has been well explored in prior works. The paper lacks deeper methodological innovations or proposals to improve model performance beyond dataset design.

The experimental section does not include a single quantitative table, making the results difficult to follow and reducing the readability and clarity of the analysis.

In Stage S3, GPT-o4-mini is used as the automatic judge for trajectory evaluation, but its rationality and reliability should be further justified through detailed case studies. This design choice may also introduce bias that favors GPT-series models.

**Questions:**

How scalable is the benchmark—could it be extended to 3D or video-based embodied data?

How consistent are human annotations across annotators, and were inter-rater reliability metrics computed?

For S3, how reliable is the GPT-o4-mini auto-scoring compared to human ratings—was any correlation analysis performed?

Have the authors examined whether S1/S2 performance correlates with actual robot execution success (e.g., using RT-1 or AgiBot replay)?

---

> ### Author Response · Authors · 2025-11-21
> **Thanks for your time to review our paper (Part 1)**
>
> Thanks for your time to review our paper and provide valuable comments. Here we will try to address some of your concerns:
>
> > The contributions are somewhat limited. Although the PIO benchmark addresses a gap in evaluating embodied reasoning through visual grounding, the concept itself has been well explored in prior works. The paper lacks deeper methodological innovations or proposals to improve model performance beyond dataset design.
>
> We appreciate the concern and would like to clarify what we see as the core technical contribution of PIO. Our goal is not to reintroduce generic visual grounding, but to embed grounding directly into embodied reasoning (ER) evaluation for VLMs, a space that is still underexplored, as most existing work assesses ER primarily through language-only outputs (for example, Gemini-ERQA). We believe our analyses provide sufficient insights, and that this benchmark can serve as an important foundation for building stronger models for embodied reasoning.
>
>
> > The experimental section does not include a single quantitative table, making the results difficult to follow and reducing the readability and clarity of the analysis.
>
> Thanks for your advice. We believe bar plots make it clearer to see which models perform better and which models perform worse, but we take the advice and put the following table results in the latest revision (marked in red) in the Appendix Table 6~7:
>
> S1/S2 score:
> *S1 and S2 subclass performance and mean scores across models.*
>
> | Model            | S1 object | S1 object part | S1 restriction | **S1 avg** | S2 afd | S2 contact | S2 predict | S2 recommend | S2 safety | **S2 avg** |
> |------------------|-----------|----------------|----------------|-----------:|--------|-----------:|-----------:|-------------:|----------:|-----------:|
> | `claude-3.7`       | 0.136     | 0.075          | 0.137          | 0.122      | 0.027  | 0.000      | 0.020      | 0.120        | 0.059     | 0.061      |
> | `gemini-2.0-flash` | 0.270     | 0.131          | 0.235          | 0.232      | 0.080  | 0.071      | 0.195      | 0.289        | 0.063     | 0.171      |
> | `gemini-2.5-flash`| 0.227     | 0.101          | 0.192          | 0.192      | 0.105  | 0.073      | 0.146      | 0.216        | 0.163     | 0.152      |
> | `gemini-2.5-pro`   | 0.334     | 0.244          | 0.267          | 0.301      | 0.130  | 0.092      | 0.261      | 0.279        | 0.193     | 0.204      |
> | `gpt-4o`           | 0.183     | 0.084          | 0.133          | 0.151      | 0.068  | 0.034      | 0.106      | 0.197        | 0.194     | 0.125      |
> | `gpt-4o-mini`      | 0.109     | 0.051          | 0.095          | 0.093      | 0.057  | 0.013      | 0.062      | 0.094        | 0.053     | 0.070      |
> | `gpt-o3`           | 0.187     | 0.122          | 0.178          | 0.170      | 0.090  | 0.050      | 0.150      | 0.225        | 0.209     | 0.151      |
> | `gpt-o4-mini`      | 0.123     | 0.077          | 0.113          | 0.111      | 0.039  | 0.001      | 0.056      | 0.162        | 0.129     | 0.089      |
> | `molmo-7b`         | 0.600     | 0.347          | 0.552          | 0.533      | 0.286  | 0.167      | 0.283      | 0.535        | 0.222     | 0.370      |
> | `qwen2.5-vl`      | 0.415     | 0.329          | 0.486          | 0.409      | 0.180  | 0.044      | 0.251      | 0.411        | 0.194     | 0.270      |
> | `robobrain`        | 0.030     | 0.037          | 0.038          | 0.033      | 0.028  | 0.028      | 0.026      | 0.058        | 0.046     | 0.040      |
> | `robopoint`        | 0.420     | 0.145          | 0.334          | 0.342      | 0.101  | 0.014      | 0.153      | 0.299        | 0.090     | 0.177      |
> | `roborefer`        | 0.687     | 0.288          | 0.605          | 0.581      | 0.183  | 0.000      | 0.333      | 0.424        | 0.417     | 0.295      |
>
>
>
>
>
>
> S3 score:
>
> | Model              | Human S3 score ↑ | GPT S3 score ↑ |
> |--------------------|------------------|----------------|
> | `gemini-2.5-pro`   | 3.91             | 3.25           |
> | `molmoact`         | 3.77             | 3.05           |
> | `gpt-o3`           | 3.20             | 2.45           |
> | `gemini-2.5-flash` | 3.00             | 2.40           |
> | `qwen2.5-vl`       | 2.65             | 2.26           |
> | `gpt-4o`           | 2.42             | 2.05           |
> | `molmo-7b`         | 1.40             | 1.48           |

---

> > ### Author Response · Authors · 2025-11-21
> > **Thanks for your time to review our paper (Part 2)**
> >
> > > In Stage S3, GPT-o4-mini is used as the automatic judge for trajectory evaluation, but its rationality and reliability should be further justified through detailed case studies. This design choice may also introduce bias that favors GPT-series models.
> >
> >
> > We put inter-rater reliability and human-vlm agreement in Appendix Section G. We assess inter-rater reliability by computing the percentage of trajectories where at least four of the five annotators agree on the same score, and the number is $82\%$ in our case. We also measure agreement between humans and VLMs: the Rank 1 (highest-ranked) model chosen by humans matches the VLM in 76\% of cases, and the Rank 5 (lowest-ranked) model matches in 84\% of cases.
> >
> > It does not favor the GPT-series models, as Gemini and Molmo-Act perform better, and the numbers are consistent with the visual results.
> >
> >
> > > Q: How scalable is the benchmark—could it be extended to 3D or video-based embodied data?
> >
> > Video is a straightforward extension, where S1–S3 become frame-wise or spatiotemporal grounding tasks with the same interface. For 3D, the main challenge is data availability rather than design; given RGB-D or simulated environments, our pointing and trajectory tasks can be lifted from 2D to 3D coordinates.
> >
> > > Q: How consistent are human annotations across annotators, and were inter-rater reliability metrics computed?
> >
> > Refer to this question ` In Stage S3, GPT-o4-mini is used as the automatic judge for trajectory evaluation, but its rationality and reliability should be further justified through detailed case studies. This design choice may also introduce bias that favors GPT-series models.`  we put results in Appendix section G.
> >
> > > Q: For S3, how reliable is the GPT-o4-mini auto-scoring compared to human ratings—was any correlation analysis performed?
> >
> > Refer to the above quesitons, we put results in Appendix section G.
> >
> > > Q: Have the authors examined whether S1/S2 performance correlates with actual robot execution success (e.g., using RT-1 or AgiBot replay)?
> >
> >
> > We focus on coarse trajectories, and we do not claim that they directly translate into executable real-world actions. However, prior work such as Hamster [a], RT-Trajectory [c], and Molmo-Act [b] shows that predicted trajectories can guide robot behavior. Based on these findings, we assume that producing reasonable trajectories is correlated with higher likelihood of successful task execution, even if the mapping is not perfectly precise.

---

### Official Review · Reviewer_Vqkf · 2025-10-31

**Soundness:** 2
**Presentation:** 2
**Contribution:** 2
**Rating:** 4
**Confidence:** 4

**Summary:**

This paper introduces an evaluation-only benchmark designed to test whether vision–language models (VLMs) can explicitly ground their reasoning in visual space. Unlike text-based embodied reasoning tasks, PIO focuses purely on spatial outputs rather than verbal answers, assessing models’ ability to predict bounding boxes, contact points, and 2D trajectories across three hierarchical stages: object localisation, task-driven grounding, and visual trace prediction.

The benchmark comprises roughly 600 human-annotated samples drawn from real-world robotic, driving, and indoor datasets, covering diverse embodied scenarios. Over ten existing VLMs—including general models such as GPT-4o, Gemini, and Claude, and grounding-specialized models such as Qwen2.5-VL and RoboRefer—are evaluated in a zero-shot setting using unified, structured prompts.

Results show that grounding-trained models perform better in localisation and affordance prediction, while general large models do relatively better in trajectory planning. No model succeeds consistently across all three stages, which the authors interpret as evidence that current VLMs remain spatially ungrounded and lack integrated embodied reasoning. PIO thus serves as a diagnostic benchmark that quantifies the gap between perception, grounding, and action reasoning.

**Strengths:**

1. The paper shifts embodied reasoning assessment from text-based QA to explicit visual grounding, offering a new angle on how well VLMs can connect perception with spatial understanding.

2. The three-stage hierarchy (referred object localization, task-level grounding, trajectory prediction) conceptually makes sense and reflects a realistic progression from perception to action, making the benchmark easy to interpret and reproducible.

3. PIO integrates data from multiple embodied domains: household, driving, and robotics, and evaluates a wide range of both general and specialized VLMs, providing a comprehensive empirical comparison.

4. The authors introduce a unified interface for structured outputs (boxes, points, trajectories), ensuring consistent evaluation across models with very different architectures and capabilities.

**Weaknesses:**

1. The benchmark evaluates all models as if they were trained to output pixel-level coordinates, masks, or trajectories, even though most general VLMs (e.g., GPT-4o, Gemini) were never optimized for such structured outputs. This conflates grounding ability with format familiarity, making cross-model comparisons unfair and potentially misleading.

2. PIO equates “embodied reasoning” with the ability to generate spatial outputs. A model that conceptually understands the task or internally attends to the correct region but cannot produce explicit coordinates is penalized as incorrect, reducing reasoning to mere coordinate prediction.

3. The benchmark never analyzes whether accurate boxes or trajectories correspond to correct semantic understanding or successful task completion, leaving unverified the claim that grounding precision reflects reasoning quality.

4. Since all models are evaluated zero-shot with a fixed prompting template, poor performance may stem from interface incompatibility rather than genuine reasoning failure. Consequently, the findings expose prompting and modality gaps more than fundamental reasoning limitations.

**Questions:**

1. Given that many general-purpose VLMs were never trained to output coordinates, masks, or trajectories, how do the authors ensure that PIO evaluates reasoning ability rather than task-format familiarity?

2. The benchmark equates embodied reasoning with spatial output accuracy. Can the authors justify this definition and explain why linguistic or internal-attention-based reasoning should be considered invalid or insufficient?

3. Have the authors analyzed whether models that produce accurate boxes or trajectories also demonstrate higher semantic or task-level correctness? Without such correlation, how is grounding quality linked to reasoning ability?

4. Do the authors believe that differences in output structure or prompt compliance might explain much of the performance gap between general and grounding-trained models? Would minimal adaptation or instruction-tuning change these results?

5. If a model correctly answers a task conceptually but fails to output precise coordinates, should this truly count as a reasoning failure? How might the benchmark be extended to capture implicit reasoning without requiring explicit coordinate prediction?

---

> ### Author Response · Authors · 2025-11-20
> **Thanks for your time to review this paper! (Part 1)**
>
> Thanks for your time to review our paper and acknowledge our contributions. Here we would like to answer some of your questions and hopefully it can address your concerns. Feel free to ask more questions and we are happy to answer:
>
> > W1:The benchmark evaluates all models as if they were trained to output pixel-level coordinates, masks, or trajectories, even though most general VLMs (e.g., GPT-4o, Gemini) were never optimized for such structured outputs. This conflates grounding ability with format familiarity, making cross-model comparisons unfair and potentially misleading.
>
>
> Grounding is an important task and is evaluated in most Vision-Language Models (also include Gemini), comparing models performance on visual grounding is widely applied in many papers (e.g. RoboPoint, RoboRefer, RoboBrain), it is definitely not a conflation of format familiarity.
>
> We do take into consideration the format e.g. MoLMO / RoboPoint are good at outputting points/ Gemini and Qwen are good at outputting bounding-boxes. And we do polish the prompt for each model to make it performs well before comparing with other models.
>
>
>
>
> > W2: PIO equates “embodied reasoning” with the ability to generate spatial outputs. A model that conceptually understands the task or internally attends to the correct region but cannot produce explicit coordinates is penalized as incorrect, reducing reasoning to mere coordinate prediction.
>
> A model that “understands” but cannot indicate the correct location is fundamentally insufficient in embodied settings. This is precisely why recent frontier models, such as Qwen-3-VL and Gemini-3, emphasize strong visual grounding capabilities in both 2D and 3D.
> Real-world embodied actions require grounded outputs: without explicit spatial grounding, a system cannot carry out meaningful physical actions, regardless of how good its internal reasoning may be. That is why we propose this benchmark, we believe it is important for both VLMs in embodied AI.
>
>
> > W3: The benchmark never analyzes whether accurate boxes or trajectories correspond to correct semantic understanding or successful task completion, leaving unverified the claim that grounding precision reflects reasoning quality.
>
> PIO intentionally **disentangles grounding from high-level semantic understanding or task completion**. Grounding itself is a standalone embodied task: the goal is to test whether a model can localize the correct object or action point, not to re-evaluate whether it semantically understands the scene. In modern VLMs, coarse semantic understanding is relatively easy and largely saturated, whereas spatial grounding remains a major bottleneck. Therefore, PIO does not require proving that accurate boxes or trajectories imply full semantic comprehension, because the benchmark is explicitly designed to measure grounding skill independently, rather than conflate it with semantic reasoning quality.
>
>
>
>
>
>
>
>
>
>
>
> > W4: Since all models are evaluated zero-shot with a fixed prompting template, poor performance may stem from interface incompatibility rather than genuine reasoning failure. Consequently, the findings expose prompting and modality gaps more than fundamental reasoning limitations.
>
>
> We carefully refine prompts for each model to ensure the interface matches the model’s expected input format. This minimizes the chance that errors come from prompt incompatibility rather than the model’s actual reasoning ability.
>
> Grounding is a core ability of MLLMs and is distinct from high-level semantic reasoning, it can be a disentangled task and is typically more difficult than semantic embodied reasoning: a model that excels at semantic reasoning may still perform poorly on grounding (e.g. Gemini-2.5-pro is not as good as Roborefer).

---

> > ### Author Response · Authors · 2025-11-20
> > **Thanks for your time to review this paper! (Part 2)**
> >
> > > Q1:
> >
> >
> > The original claim that `many general-purpose VLMs were never trained to output coordinates, masks, or trajectories `  is not accurate. In reality, most modern general VLMs include grounded data as an important part of their training. For open-source models, grounding has been present since early systems; for example, LLaVA [b] includes bounding-box prediction in its SFT stage. Newer models such as Qwen-2.5-VL, InternVL [c], and MoLMO incorporate bounding boxes or point-level grounding extensively during pre-training. For closed-source models, grounding is also emphasized. Many proprietary VLMs highlight grounding as a key capability at release, for example Gemini [a].
> > In our experiments, GPT-4o and Claude perform worse than other models, although they could still produce reasonable coordinates because of their strong general reasoning ability. We do not believe this reduces the value of our task. It remains a meaningful evaluation of how well models handle structured grounding outputs in pixel space, and the task itself has been  well recognized [a, d].
> >
> >
> > > Q2:
> >
> >
> > Grounded outputs are essential for VLMs to interact with the physical world, since language-only responses cannot directly specify actions or locations.
> >
> >
> > > Q3:
> >
> >
> >
> > First, grounding quality itself is linked to reasoning ability because producing a correct box or trajectory already requires several non-trivial reasoning steps. To output the right coordinates, the model must first understand the instruction, identify the correct object or region in the scene, resolve spatial relations such as “left of” or “in front of,” and then map that understanding to a concrete location. If any of these reasoning steps fails, the grounding output is usually wrong.
> >
> > Second, we do not necessarily need a strong global correlation between grounding accuracy and task-level success for our benchmark to be meaningful. Our goal is to isolate and evaluate the grounding component itself, similar to how object detection benchmarks evaluate localization without requiring a direct correlation to image captioning quality. Grounding and high-level semantics are complementary abilities: a model may answer semantic questions reasonably well while still failing to localize the right object, and revealing this gap is precisely the purpose of our benchmark.
> >
> >
> >
> >
> > > Q4:
> >
> > We agree that models like GPT-4o and Claude are not heavily trained on visual grounding data and could likely improve with additional fine-tuning or instruction-tuning. However, we do not think this weakens the purpose of our benchmark. A benchmark is meant to reveal that some models currently perform well and some perform poorly on a given ability, regardless of whether future fine-tuning could close the gap.
> >
> >
> >
> >
> >
> > > Q5:
> >
> >
> >
> >
> > First, we agree that failing to output precise coordinates **does not** mean the model has no conceptual understanding. A model can often understand that it should act on "the mug on the right" in a loose, semantic sense, yet still miss the exact pixel location. In our benchmark, we deliberately count this as a grounding failure rather than a pure semantic reasoning failure. The reason is that for embodied agents, conceptual understanding is only useful if it can be turned into a concrete spatial target that a robot or controller can act on. If the coordinates or trajectory are wrong, the task still fails in practice, even if the model's internal reasoning was partially correct. Our claim is therefore not that the model cannot reason, but that its reasoning cannot be reliably operationalized in the physical world.
> >
> > Second, we agree that it is valuable to capture implicit reasoning beyond strict coordinate prediction, and our benchmark could be naturally extended in that direction. For example, one could add: "semantic correctness" track where the model identifies the correct object or region via multiple choice, or answers which object should be acted on in language, without any coordinates, the extensions would help disentangle conceptual understanding from low-level spatial precision. Our current work focuses on the latter, that is, on whether models can produce actionable grounded outputs. We see evaluating implicit reasoning as a complementary future direction rather than a contradiction to the current benchmark.
> >
> > Please let me know if you have further concerns, we are happy to answer your questions.\
> >
> > [a] https://ai.google.dev/gemini-api/docs/image-understanding
> > [b] Visual instruction tuning
> > [c] Internvl: Scaling up vision foundation models and aligning for generic visual-linguistic tasks
> > [d] Qwen3-VL: Sharper Vision, Deeper Thought, Broader Action

---

### Official Review · Reviewer_XKnN · 2025-10-31

**Soundness:** 1
**Presentation:** 2
**Contribution:** 1
**Rating:** 2
**Confidence:** 4

**Summary:**

This work proposes a new benchmark for evaluating the visual grounding ability of vision-language models (VLMs) in embodied AI tasks. The benchmark contains approximately 600 QA pairs, categorized into three types: referred-object localization, task-driven pointing, and visual trace prediction across diverse scenarios. It further evaluates both open-source and closed-source MLLMs, offering valuable insights derived from the comparative results.

**Strengths:**

This work focuses on visual grounding in images, providing an interpretable way to compare the visual understanding capabilities of different VLMs.

It designs three task types that consider the use of VLMs as an interface for low-level policy learning.

**Weaknesses:**

The proposed tasks including referred-object localization, task-driven pointing, and visual trace prediction have already been explored in prior work, making the task-level novelty of the benchmark relatively limited.

The overall work is engineering-oriented, focusing primarily on data curation rather than introducing new methods to enhance MLLM performance, which reduces the novelty and conceptual contribution of the paper.

The tasks span household rooms, kitchen environments, driving scenes, and robotic manipulation scenarios, which results in some domain overlap. Moreover, the motivation for including driving scenes is unclear and appears loosely connected to embodied reasoning.

**Questions:**

Would it be possible to conduct a systematic evaluation of embodied agents across multiple dimensions  including referred-object localization, task-driven pointing, and visual trace prediction by combining existing benchmarks instead of creating a new one?

It would be helpful to briefly discuss the cost and feasibility of performing benchmarking experiments on PIO.

Could you clarify the motivation for dividing the scenarios into household rooms, kitchen environments, driving scenes, and robotic manipulation settings?

How many different types of robots, object categories, and object properties (e.g., color, material) are included in this benchmark? Providing such statistical details would strengthen the clarity and completeness of the work.

---

> ### Author Response · Authors · 2025-11-19
> **Thanks for your time to review our paper! (Part 1)**
>
> Thanks for your time to review our paper and valuable questions you raised, we answer the questions below and hopefully it will address some of your concerns:
>
> > W1: The proposed tasks including referred-object localization, task-driven pointing, and visual trace prediction have already been explored in prior work, making the task-level novelty of the benchmark relatively limited.
>
> We would like to clarify that our contribution does **not** lie in proposing entirely new tasks. Instead, we systematically construct an **evaluation pipeline** to assess embodied reasoning in complex, real-world embodied scenes. To the best of our knowledge, our benchmark is the **first** to jointly cover *referred-object localization*, *task-driven grounding/pointing*, and *visual trace generation*, and to provide **standardized evaluation metrics** for all three within a unified embodied environment (Table 1).
>
> Existing grounding benchmark e.g., RefSpatial (Zhou et al., 2025) focus primarily on object localization and free-space placement. In contrast, our benchmark spans **richer categories**, including object parts, affordances, recommendations, constraints, contacts, and safety considerations. This broader taxonomy enables a **much deeper assessment of embodied reasoning** than prior task-specific benchmarks.
>
> ---
>
>
> > W2: The overall work is engineering-oriented, focusing primarily on data curation rather than introducing new methods to enhance MLLM performance, which reduces the novelty and conceptual contribution of the paper.
>
>
> We respectfully disagree that the contribution is primarily engineering-oriented. Our goal is not to propose a new MLLM method, but to address a critical missing component in current research: a systematic, diagnostically meaningful evaluation framework for embodied reasoning.
>
> Existing evaluations for embodied reasoning of MLLMs overwhelmingly measure image-level or language-level abilities, while visual grounding in embodied reasoning scenarios remains largely unassessed due to the absence of proper benchmarks.
>
> ---
>
> > W3: The tasks span household rooms, kitchen environments, driving scenes, and robotic manipulation scenarios, which results in some domain overlap. Moreover, the motivation for including driving scenes is unclear and appears loosely connected to embodied reasoning.
>
>
> Each scene type serves a distinct purpose: kitchen environments focus on cooking-related interactions; driving scenes evaluate reasoning in diverse road conditions; robotic manipulation centers on tabletop arm control; and household scenes capture broader indoor activities.
> These domains are sourced from different base datasets, ensuring that content overlap is minimal while providing complementary embodied reasoning challenges.
>
> Also, we believe autonomous-driving is an important part of embodied AI [a], it also requires perception, action prediction and spatial reasoning in physical world.
>
> ---
>
> > Q1: Would it be possible to conduct a systematic evaluation of embodied agents across multiple dimensions including referred-object localization, task-driven pointing, and visual trace prediction by combining existing benchmarks instead of creating a new one?
>
>
> The main challenge is that most large referred-object localization dataset e.g. RefCoco does not focus on embodied scenarios. Some in-domain visual-grounding dataset e.g. Where2Place and RefSpatial focus only on indoor table scenarios and are always small (~100 examples). Some in-domain embodied reasoning dataset e.g. Gemini-ERQA does not include visual grounded annotations.
>
> For visual tracing there are no existing dataset used to systematically evaluate this ability.
>
> ---
>
> > Q2: It would be helpful to briefly discuss the cost and feasibility of performing benchmarking experiments on PIO.
>
>
> Thanks for pointing it out. Below we provide the time and GPU memory cost for running the full evaluation on our benchmark using **Qwen2.5-VL-8B** as an example.
>
> | Model         | GPU Type      | GPU Memory | Tasks Evaluated | Runtime     |
> | ------------- | ------------- | ---------- | --------------- | ----------- |
> | Qwen2.5-VL-8B | A6000 (48 GB) | 48 GB      | S1 + S2         | ~1.5 hours  |
> | Qwen2.5-VL-8B | A6000 (48 GB) | 48 GB      | S3              | ~30 minutes |
>
> Total runtime for the complete benchmark (S1+S2+S3) on a single A6000 is **approximately 2 hours**.
>
> ---
>
>
>
> [a] What Is Embodied AI? https://www.nvidia.com/en-eu/glossary/embodied-ai/
>
> [b] RoboCasa: Large-Scale Simulation of Everyday Tasks for Generalist Robots
>
> [c] Behavior robot suite: Streamlining real-world whole-body manipulation for everyday household activities

---

> ### Author Response · Authors · 2025-11-19
> **Thanks for your time to review our paper (Part 2)**
>
> > Q3: Could you clarify the motivation for dividing the scenarios into household rooms, kitchen environments, driving scenes, and robotic manipulation settings?
>
> Different scene categories target different aspects of embodiment. We draw inspiration from recent simulation pipelines in robotic manipulation such as RoboCasa [b], which emphasize kitchen environments as rich, affordance-dense embodied settings. Our household scenes are motivated by humanoid-agent datasets (e.g., BEHAVIOR-robot-suite [c]), where agents navigate and interact across entire rooms. In contrast, on-table manipulation scenes focus on fixed-base robot arms, highlighting contact reasoning and fine-grained object interactions. And finally auto-driving focuses on road scenarios where the car is the embodiment.
>
> ---
>
>
> > Q4: How many different types of robots, object categories, and object properties (e.g., color, material) are included in this benchmark? Providing such statistical details would strengthen the clarity and completeness of the work.
>
> That’s a good point. For end-effector we have robot arm, humanoid, car, and also human hands. We did statistics of object categories, colors, locations, materials and task types as below, which will also be put into the Appendix in the next revision:
>
> | Category              | Count                  | Notes                                                   |
> |-----------------------|------------------------|---------------------------------------------------------|
> | Different objects | **~185**           | gripper, cup, glass cup, candy box, chip can, Coke, corn, mushrooms, cucumbers, carrot, potato, tray, key...          |
> | Colors           | **8**                  | red, black, white, blue, green, orange, silver-gray, grey |
> | Locations        | **70–90**             | left side, right side, on the table, on the stove, pedestrian lane, opposite lane, middle shelf, top drawer, bottom drawer, upper-left stove, lower-right stove, walkway, bridge, parking lot, top tray, bottom drawer ...             |
> | Materials         | **3–4 explicit**, ~7 total | wooden, glass, silver-gray, metal ...|
> | Task types        | **~25–30**             | pick-place/pour/open-close/swith lane/merge lane/ ...    |
>
> ---
>
> [a] What Is Embodied AI? https://www.nvidia.com/en-eu/glossary/embodied-ai/
>
> [b] RoboCasa: Large-Scale Simulation of Everyday Tasks for Generalist Robots
>
> [c] Behavior robot suite: Streamlining real-world whole-body manipulation for everyday household activities

---

### Author Response · Authors · 2025-12-01
**Author Final Response**

Thanks to the AC and all reviewers for taking the time to evaluate our paper. Due to the unexpected issues in the review process this year, we unfortunately did not have sufficient opportunity to discuss with the reviewers. Here we provide a final response to summarize.

Our contribution is a meaningful and timely step toward **evaluating embodied reasoning via explicit visual grounding**, rather than only language or MCQ outputs.

### (1) Why the contribution matters

* PIO is **not just another grounding dataset**: it is, to our knowledge, the first benchmark that **jointly and systematically evaluates three embodied grounding stages**—referred-object localization (S1), task-driven grounding (S2), and visual trace prediction (S3)—in realistic household, kitchen, driving, and tabletop robotic scenes with a unified interface and metrics.
* It targets a **missing evaluation axis**: existing embodied reasoning work typically tests only language answers, while modern VLMs are increasingly expected to output **grounded spatial signals** (points, boxes, trajectories) for real-world control.

### (2) How we addressed key concerns

* **Novelty / “engineering-only”**: We clarified that the core novelty lies in **embedding grounding directly into ER evaluation** with a rich taxonomy (objects, parts, affordances, safety, etc.), not inventing new task primitives.
* **Fairness and “format familiarity”**: We noted that most modern VLMs *are* trained with grounded data, followed visual-grounding practice from RoboRefer/RoboPoint, and **carefully tuned prompts per model** to reduce interface bias.
* **Supplementary Results**: We added quantitative tables for all stages in the **revision**. Also, we add inter-rater analysis in **revision** to support the reliablity of our S3 evaluation.


Given the above, we believe PIO is a meaningful, well-motivated benchmark contribution that fills an important gap in embodied VLM evaluation and is closer to the positive reviews than to the single negative one.

---

### Meta-Review · Area_Chair_gJ1u · 2026-01-06

**Summary:**

The paper proposes "Point-It-Out" (PIO), a benchmark evaluating Vision-Language Models (VLMs) on embodied reasoning via visual grounding across three stages (S1: localization, S2: task-driven pointing, S3: visual trace prediction). While Reviewer dRu3 championed the work for moving beyond Multiple-Choice Questions (MCQs), the consensus among other reviewers (XKnN, Vqkf, Gmue) leans towards rejection. The primary concerns center on the limited novelty of the proposed tasks, the potential conflation of "reasoning" with "formatting familiarity" (coordinate output), and the lack of validation linking grounding metrics to actual task execution success.

**Reviewer Concerns:**

Addressed:

1）The authors improved the manuscript by adding quantitative tables and inter-rater reliability analysis for Stage 3 (S3) in response to Reviewers Gmue and dRu3.

2）Clarifications regarding the domain distribution and prompt tuning were provided.

Outstanding:

1）Novelty (Crucial): As noted by Reviewer XKnN and the AC, the individual tasks are not novel. Specifically, visual trace prediction (S3) has been extensively explored in prior works like RoboBrain, limiting the innovative value of this benchmark. The contribution is viewed primarily as engineering-oriented data curation rather than methodological or conceptual advancement.

2）Validity of Evaluation: Reviewer Vqkf raised a significant concern that the benchmark may unfairly penalize general-purpose VLMs (like GPT-4o) for format incompatibility (outputting coordinates) rather than genuine reasoning failures. The rebuttal did not fully alleviate concerns that the benchmark conflates output formatting with semantic understanding.

3）Correlation with Reality: There remains a gap in justifying whether high scores on PIO actually translate to success in real-world robotic execution.

**Reviewer Scores:**

Reviewer XKnN (2): Unlikely to change. The fundamental concern about task-level novelty remains valid.

Reviewer Vqkf (4): Likely to remain unchanged. The philosophical disagreement on whether "grounding accuracy equals reasoning ability" was not resolved.

Reviewer Gmue (4): May marginally improve due to added stats, but likely stays borderline due to limited contribution.

Reviewer dRu3 (8): Likely to remain positive, valuing the systematic categorization.

---

### Decision · Program_Chairs · 2026-01-26

Reject